# Optimization of Protocols for the Induction of Callus and Plant Regeneration in White Clover (*Trifolium repens* L.)

**DOI:** 10.3390/ijms241411260

**Published:** 2023-07-09

**Authors:** Tiangang Qi, Tao Tang, Qinyu Zhou, Weiqiang Yang, Muhammad Jawad Hassan, Bizhen Cheng, Gang Nie, Zhou Li, Yan Peng

**Affiliations:** College of Grassland Science and Technology, Sichuan Agricultural University, Chengdu 611130, China; 17863660603@163.com (T.Q.); shmilyxxx@126.com (T.T.); 19982939311@163.com (Q.Z.); weiqiang0214@163.com (W.Y.); jawadhassan3146@gmail.com (M.J.H.); Chengbizhengrass@163.com (B.C.); nieg17@sicau.edu.cn (G.N.); lizhou1986814@163.com (Z.L.)

**Keywords:** white clover, leaf, callus, antioxidants, hormones, *Agrobacterium tumefaciens*

## Abstract

White clover is a widely grown temperate legume forage with high nutritional value. Research on the functional genomics of white clover requires a stable and efficient transformation system. In this study, we successfully induced calluses from the cotyledons and leaves of 10 different white clover varieties. The results showed that the callus formation rate in the cotyledons did not vary significantly among the varieties, but the highest callus formation rate was observed in ‘Koala’ leaves. Subsequently, different concentrations of antioxidants and hormones were tested on the browning rate and differentiation ability of the calluses, respectively. The results showed that the browning rate was the lowest on MS supplemented with 20 mg L^−1^ AgNO_3_ and 25 mg L^−1^ VC, respectively, and the differentiation rate was highest on MS supplemented with 1 mg L^−1^ 6-BA, 1 mg L^−1^ KT and 0.5 mg L^−1^ NAA. In addition, the transformation system for *Agrobacterium tumefaciens*-mediated transformation of 4-day-old leaves was optimized to some extent and obtained a positive callus rate of 8.9% using green fluorescent protein (GFP) as a marker gene. According to our data, by following this optimized protocol, the transformation efficiency could reach 2.38%. The results of this study will provide the foundation for regenerating multiple transgenic white clover from a single genetic background.

## 1. Introduction

White clover (*Trifolium repens* L.) is one of the most extensively grown perennial leguminous forage grasses in the world. It is native to Europe and is widely distributed in temperate and subtropical high-altitude regions [1]. It is estimated that 3–4 million hectares of white clover are planted annually worldwide [2]. White clover is often mixed with companion grasses, such as perennial ryegrass and orchardgrass, because of its strong nitrogen fixation capacity and nutrient enrichment [3,4,5]. It also plays a crucial role in improving pasture quality, thereby providing the basis for global livestock production [6,7,8]. However, white clover has a low tolerance to drought and salinity, which may be an important factor limiting its development [9,10,11]. Therefore, breeding high-quality white clover varieties with wide adaptability has become particularly urgent and important.

Generating new varieties of white clover with superior stress tolerance traits by conventional breeding methods is a long and arduous process, as both drought and salt tolerance in plants are quantitative traits controlled by multiple genes [12,13]. In contrast, transgenic technology is expected to allow additional genetic gain in white clover by introducing superior traits [14,15]. In recent years, important genomic resources of white clover have been effectively utilized [16,17,18]. Although transformation protocols for white clover are readily available, the ability to exploit these resources is limited by self-incompatibility, heterozygosity, heteroploidy, and high levels of genetic heterozygosity, which can eventually lead to loss of heterozygosity and inbreeding decline [13,19,20]. As a result, even seeds harvested from the same plant have widely varying genotypes.

White clover is a stolon plant that can be easily propagated by cuttings from mature or growing plants. These cuttings can provide explant material that is similar to seedlings [21,22]. Previous studies have reported successful vegetative propagation of white clover by callus induction from leaf explants [23]. Callus can be induced from various white clover explants, such as leaves [23,24], petioles [25], hypocotyls [26], stems [27], and roots [28]. Moreover, some reports have found that immature conidial embryos can produce asexual plants [21]. These findings provide strong prerequisites for the cultivation of white clover. However, plant transformation and regeneration are highly dependent on the species and genotype, as there are significant genetic differences in individual cell among different species and different genotypes of the same species [29,30,31,32]. Therefore, the selection of suitable explants is crucial for a single genetic background.

*Agrobacterium tumefaciens*-mediated methods have been widely used for plant genetic transformation in recent years [33,34,35,36]. As early as 1987, White established a transformation system to obtain the first transgenic white clover plants with *Agrobacterium*-mediated methods [27], but the frequency of transformation protocols was extremely low (<1%). Subsequent studies on this species reported *Agrobacterium*-mediated transformation efficiencies ranging from 0.3% to 10% [37,38,39,40]. To transform a genetically diverse species such as white clover, a genotype-independent regeneration scheme is required. For this purpose, this study utilized the more readily available leaves as explants to induce calluses and their regeneration. This regeneration was significantly less genotype-dependent compared to previous reports, which may represent a means to improve white clover using *Agrobacterium tumefaciens*-mediated transformative regeneration.

In our previous study, we successfully obtained white clover plants from calluses, realizing a protocol for *Agrobacterium*-mediated cotyledon and root transformation. In this study, we evaluated the leaf callus induction rate in different genotypes of white clover and identified the cultivar ‘Koala’ as a suitable candidate. We also systematically investigated the effects of different antioxidant and hormone concentrations on the callus induction and differentiation abilities, aiming to obtain a more suitable type of leaf callus for embryogenic differentiation. This study provides a technical basis for establishing an efficient regeneration system in white clover.

## 2. Results

### 2.1. Formation of Calluses on White Clover

The highest callus induction rate was used as an evaluation index to screen the white clover varieties (Figure 1). The results show that the callus induction rate of the cotyledon explants of all 10 varieties reached more than 85% (Appendix A; Figure 2a). Interestingly, throughout the 20 days of callus induction, there was no significant difference in the induction efficiency of the calluses among the different varieties of white clover cotyledon sources (Figure 2a). All varieties of leaf explants showed more than 60% induction of calluses on CCM, and the highest and lowest callus induction rates were observed in ‘Koala’ and ‘Haifa’ white clover leaf explants, respectively (Figure 2b). Therefore, ‘Koala’ was chosen for the subsequent experiments.

### 2.2. Effect of Antioxidant on Callus Browning

Browning is common during the callus induction process because of the relatively high content of phenolics in white clover tissues. Therefore, we added AgNO_3_ and VC into the induction medium. As shown in Figure 3a, the browning rate of the embryonic calluses was much higher without AgNO_3_ than in other concentration groups. The lowest browning rate (45%) was observed when CCM containing 20 mg L^−1^ AgNO_3_ was added (Figure 3a and Figure 4b). In addition, the browning rate of the embryonic calluses gradually increased with increasing AgNO_3_ concentrations.

As previously described for AgNO_3_, VC had a similar effect in reducing the browning rate of calluses. The lowest browning rate of embryonic calluses (32%) was observed in CCM medium supplemented with 25 mg L^−1^ VC compared to the control and even better than AgNO_3_ (Figure 3b and Figure 4c). Similarly, the browning rate of embryonic calluses on medium containing 30 mg L^−1^ VC started to increase as the VC concentration increased. In addition, the antioxidant VC (25 and 30 mg L^−1^) was more effective than AgNO_3_ in reducing the rate of browning of the callus at the same concentration gradient (Appendix A).

### 2.3. Effect of Hormone Concentrations on the Redifferentiation of White Clover Calluses

Based on a previous study, we determined 1 mg L^−1^ 6-BA in CCM as the optimum concentration [41]. The results show that different combinations of 1 mg L^−1^ 6-BA and NAA both induced the regeneration of calluses to differentiate clumped shoots, and some of the calluses gradually turned brown from the periphery to the middle without further differentiation (Figure 4d,e). Even so, more than 50% of the calluses regenerated shoots at different concentrations of NAA compared to 75% of the regenerated shoots at 0.25 mg L^−1^ NAA (Figure 5a). Likewise, we transferred embryonic calluses to CDM2 medium supplemented with 0.1 mg L^−1^ NAA and different concentrations of KT and cultured them using the same conditions described above. More than 50% of the calluses regenerated shoots under different concentrations of KT, while the percentage of regenerated shoots under 1 mg L^−1^ KT conditions was the highest (70%) (Figure 5b).

We selected three hormone combinations based on their superior effect on callus differentiation to examine their ability to induce differentiation. The highest differentiation rate (82.5%) was observed when CDM containing 1 mg L^−1^ 6-BA, 1 mg L^−1^ KT and 0.5 mg L^−1^ NAA was added (Figure 5c). In addition, regenerated shoots obtained from healthy callus differentiation were larger and greener than the two hormone combinations (6-BA and NAA; KT and NAA) (Figure 4f). These results suggest that the cocombination of 6-BA, KT and NAA significantly improved the differentiation of the calluses.

### 2.4. Agrobacterium tumefaciens-Mediated Transformation

We used an *Agrobacterium*-mediated approach to transform *T. repens* ‘Koala’ leaves using AS and Blp antibiotics. The explants were cocultured on CM for 4 days, followed by induction of callus on CIM (Figure 6a–d). We obtained light-green, solid calluses that were able to grow in CCM-S medium and were resistant to Blp antibiotic screening (Figure 6f). From these resistant calluses, we regenerated multiple green shoots and transferred them to CRM-S medium for continued growth (Figure 6h). To avoid *Agrobacterium* overgrowth leading to necrosis of the callus and reduced transformation efficiency, we checked the cultures regularly. When the explant was transferred into CCM-S for approximately 60 days, we identified positive transformants using fluorescence microscopy. Positive transformants could be identified by their bright green fluorescence (Figure 6e). The results show that we successfully established the genetic transformation system of white clover leaves as explants.

### 2.5. Identification of Transgenic Plants

After obtaining the transgenic plants, we verified positive plants with PCR using vector-specific primers (Figure 7). Finally, we calculated the positive transformants in the transgenic experiments, and the results showed that the transformation efficiency was 2.38% (5 transgenic plants/210 explants) (Appendix A).

## 3. Discussion

The majority of previous studies on white clover regeneration have used cotyledons and mature or immature embryos as explants [21,40]. Both the induction of calluses from explants and the induction of embryonic calluses are dependent on the action of hormones. However, using different organs of the same plant as explant materials resulted in large differences in the callus initiation efficiency under the same type and concentration of hormones [41]. In a previous study, we obtained culture medium compositions (CCMs) that were more suitable for callus induction in white clover explants [42]. In this study, cotyledons and leaves of different varieties of white clover were used as explants for the induction of calluses. Interestingly, the induction efficiency of cotyledon explants did not differ significantly among varieties; however, there were large differences in the induction efficiency of leaf explants (Figure 2). This phenomenon may be related to the organ structure, differentiation status and dedifferentiation ability of plant explants.

Ideally, the introduced target gene should be expressed in each genotype of the transgenic variety. This could be achieved by keeping the transgene in a pure state, thus ensuring the uniformity of the genetic background of the transgene [31,32]. Leaf explants are well-known to be the best resources for efficient regeneration from callus cultures of alfalfa [43]. In this study, cotyledon explants had a higher overall induction efficiency than leaf explants in white clover, because white clover is an obligate outcrossing plant and each seed represents a distinct genotype [15]. Therefore, plants regenerated from cotyledons may lose target genes during transgenic progeny development. Hence, we selected leaves (*T. repens* cv. Koala) as explants to induce callus and regeneration to ensure a single genetic background.

Browning is the key factor determining the success of plant tissue culture, and its occurrence is usually dependent on plant species and genotype (Figure 5b,c). Plants with severe browning often contain relatively high levels of phenolic compounds and can affect the viability and regeneration potential of callus and, thus, need to be effectively controlled for successful plant tissue culture [44,45,46]. In fact, antioxidants have been widely used to inhibit browning in many plant species, including *Zea mays* [46], *Abelmoschus esculentus* [47] and *Brassica oleracea* [48]. In this study, we found that 20 mg L^−1^ AgNO_3_ and 25 mg L^−1^ VC reduced the browning rates of calluses at three weeks of induction (Figure 3). Moreover, VC was more effective than AgNO_3_ in preventing browning of callus (Appendix A). However, higher concentrations of AgNO_3_ or VC increased the browning rate, indicating that the optimal concentration of antioxidants depends on the explant type.

It is well known that 2,4-D is an important hormone for the induction of somatic embryogenesis and is usually removed after dedifferentiation in order to prevent further differentiation of spherical embryos into secondary embryos [49]. Moreover, some researchers reported that NAA and 6-BA can be used together to promote calli induction and plant regeneration in some species [50]. However, the precise regulatory mechanism of tissue culture has not yet been fully revealed. A viewpoint that is generally accepted is that callus induction and organ differentiation are mainly determined by hormone balance [51,52]. In this study, the differentiation rates for the two combinations (1 mg L^−1^ 6-BA and different concentrations of NAA or 0.1 mg L^−1^ NAA and different concentrations of KT) were 75% and 70%, respectively (Figure 4a,b). Considering the combined use of hormones, we selected the combination of 1 mg L^−1^ 6-BA with 1 mg L^−1^ KT and different concentrations of NAA to optimize the differentiation efficiency of the callus. The results revealed an overall increase in the differentiation rate compared to the previous combination of the two hormones, with the highest differentiation rate of 82.5% (Figure 4c). In addition, a further increase in KT improved the quality of callus differentiation. However, a decrease in the callus differentiation rate occurred with a gradual increase in the NAA or KT concentration. A possible reason for this might be connected with the hormone levels, as higher hormone levels interfere with the development of somatic embryos and disrupt the normal rearrangement of chromatin structures in plant cells [41,53].

The establishment of *Agrobacterium tumefaciens*-mediated transformation protocols depends heavily on the availability of explants, and various culture conditions influence the induction and differentiation of calluses. However, white clover leaves are extremely challenging to transform because of their high endophytic fungal load that causes contamination and explant mortality [36]. Previous studies have used cotyledons as explants and reported regeneration rates of resistant shoots (5.45% and 4.15%, respectively) [42]. To overcome these limitations, we optimized the culture conditions to reduce contamination and enhanced callus induction and differentiation (Figure 6). At the same time, we selected the more stringent glufosinate (25 mg L^−1^) to suppress the excessive growth of A. tumefaciens on the callus surface [35,54]. Moreover, to allow for visually monitoring transformation events, we transfected leaf explants with *Agrobacterium tumefaciens* GV3101, which contained the pCAMBIA 3300 binary vector, as well as the GFP reporter gene, to allow for visual monitoring [55]. Finally, we obtained transgenic plants with a single genetic background. Although the transformation efficiency was only 2.38%, the development of the protocol provides a feasible technical basis for white clover molecular breeding.

## 4. Materials and Methods

### 4.1. Plant Material

Different white clover (*Trifolium repens* L.) varieties (Appendix A) were used for regeneration and transformation. Seeds were sterilized initially with ethanol (70%, *v*/*v*) for 5 min, then in HgCl_2_ solution (0.1%, *w*/*v*) for 10 min and washed five times with sterile distilled water. Subsequently, approximately 200 seeds were scattered on sterile filter paper (10 × 10 cm) for germination for 4 days. The germinated Petri dishes (9 × 9 cm) were placed at a 24/22 °C circadian temperature with 16 h/8 h light/dark cycle and light intensity of 150 μmolm^−2^ s^−1^. One-month-old leaves of different white clover varieties were taken and disinfected with ethanol (70%, *v*/*v*) for 2 min, then disinfected in HgCl_2_ solution (0.1%, *w*/*v*) for 5 min and rinsed five times with sterile distilled water. Plants were maintained in controlled conditions, i.e., 24/22 °C day/night temperature, 16 h/8 h light/dark cycle and light intensity of 150 μmol m^−2^ s^−1^.

### 4.2. Callus Induction

Based on a previous study, we designed a white clover callus induction medium formulation (Appendix A). Ten different varieties of white clover cotyledons and leaves were used as explants to investigate the effect of the genotype and explant type on the callus induction. Subsequently, the cotyledons were divided into 2–3 sections and individual leaves were cut into 5–6 parts to serve as explants for the induction of callus. The explants were placed on Petri dishes containing Murashige and Skoog (MS) medium supplemented with 0.5 mg L^−1^ 6-BA, 2 mg L^−1^ 2,4-D and 30 g L^−1^ sucrose (referred to as clover callus medium, CCM). In addition, the pH of the medium was adjusted to 5.8 before autoclaving at 121 °C. The entire callus induction process was carried out in a dark environment, and the embryonic callus induction rate of different varieties was counted after the explants were cultured in the dark for 20 days. Three independent experimental replicates (70 cotyledon explants and 20 leaf explants per Petri dish) were performed for each medium and explant. The callus induction rate = (number of explants that formed a callus/the total number of inoculated explants in each treatment) × 100%.

### 4.3. Antioxidant Concentration Test

To investigate the effect of the antioxidant concentration on the browning rate of the calluses, the leaves of the abovementioned variety ‘Koala’ were used as explants to induce calluses and then inoculated with CCM containing different concentrations of silver nitrate (AgNO_3_; 0, 5, 10, 15, 20, 25 and 30 mg L^−1^). The same experiment was performed for vitamin C concentrations, with the addition of vitamin C (VC; 2.5, 5, 7.5, 10, 15 and 20 mg L^−1^) to the CCM and no VC set as a control. Three independent experimental replicates (40 tissues per set of experiments) were performed for each oxidant concentration. Calluses from both treatments were kept at 22 °C in the dark, and the medium was refreshed after every 20 days. The callus browning rate was measured on the 25th day of subculturing. The browning rate = (number of browning explants/the total number of inoculated explants in each treatment) × 100%. 

### 4.4. Effect of Hormones on the Differentiation of Calluses

The effect of different concentrations of different plant hormones on the differentiation of embryonic calluses was tested. 50-day-old fresh embryogenic callus differentiation was tested on clover differentiation medium (CDM1) (Appendix A) supplemented with 6-BA (1 mg L^−1^) and disparate quantities of naphthalene acetic acid (NAA; 0.025, 0.05, 0.1, 0.25, 0.5, 0.75 and 1 mg L^−1^) as well as on CDM2 supplemented with NAA (1 mg L^−1^) and disparate quantities of kinetin (KT; 0.5, 0.75, 1, 1.5 and 2 mg L^−1^) at a culture temperature of 23 °C and a light/dark cycle of 16 h/8 h. To analyze the combined effect of hormones, 1 mg L^−1^ KT and different quantities of NAA were added to CDM1. After 25 days of incubation, the differentiation rates of embryonic calluses at different concentrations were counted and verified with three replicates of approximately 40 calluses for each concentration.

### 4.5. Agrobacterium tumefaciens-Mediated Leaf Transformation

The pCAMBIA 3300 plasmid, which contained a GFP reporter gene driven by a super promoter and a glufosinate resistance gene, was used in this experiment for the transformation of somatic embryos. Appendix A shows the composition of the medium for the induction and proliferation of white clover callus. Mature leaf explants of *T. repens* ‘Koala’ were immersed in liquid clover infection mediums (CIM) supplemented with *Agrobacterium tumefaciens* (OD600 = 0.5) and incubated at 28 °C for 20–25 min at 80–100 rpm. Subsequently, the explants were dried on sterile filter paper and cocultured with *Agrobacterium tumefaciens* on the cocultivation medium (CM) for 4 days at 23 °C. After washing 3–4 times with sterile water, the explants were transferred to clover callus selection medium (CCM-S) supplemented with 300 mg L^−1^ cefotaxime (Cef) and 25 mg L^−1^ glufosinate (Blp) and incubated at 23 °C. The selection was implemented in the dark for 50–60 days with media alteration every 20 days. Then, healthy calluses were placed on CDM-S medium containing 25 mg L^−1^ Blp and induced by differentiation at 23 ℃ on a 16h/8h light/dark cycle. After 50–60 days, regenerated green shoots under Blp selection were placed on CRM selection medium (CRM-S) at 23 °C on a 16 h/8 h light/dark cycle for whole-plant development.

### 4.6. Identification of Transformed Explants and Plants

After 50–60 days of incubation of the infiltrated explants, the GFP fluorescence of the positive calluses was observed using a GelView 6000Plus multicolor fluorescence imaging system with a fluorescence module. Genomic DNA was extracted from the leaves of transgenic plants surviving under Blp selection and verified with PCR amplification using vector-specific primers F (5′-ATGGTGAGCAAGGGCGAGGAG-3′) and R (5′-TCAAAGATCTACCATGTACAGCTCGT-3′).

### 4.7. Statistical Analysis

Statistical analyses were conducted using SPSS version 22 software (SPSS Inc., Chicago, IL, USA). Significant differences between treatments were tested at the 0.05 level of probability following a one-way ANOVA combined with Fischer’s least significant difference (LSD) test. In most cases, Tukey tests were performed considering the different number of healed wounds in each treatment.

## 5. Conclusions

Based on previous studies, we demonstrated the feasibility of using different antioxidant concentrations and different hormone concentrations in *T. repens* ‘Koala’ leaves for callus culture conditions, redifferentiation and *Agrobacterium*-mediated genetic transformation. In this study, the effects of AgNO_3_ and VC concentrations and different amounts of hormone combinations on the browning and differentiation abilities of the calluses were evaluated, respectively. In addition, *Agrobacterium*-mediated genetic transformation under new and optimized protocol conditions successfully obtained transgenic plants, with a transformation efficiency of 2.38%. We believe that our protocol will facilitate the study of white clover traits and contribute to the future molecular breeding of white clover.

## Figures and Tables

**Figure 1 ijms-24-11260-f001:**
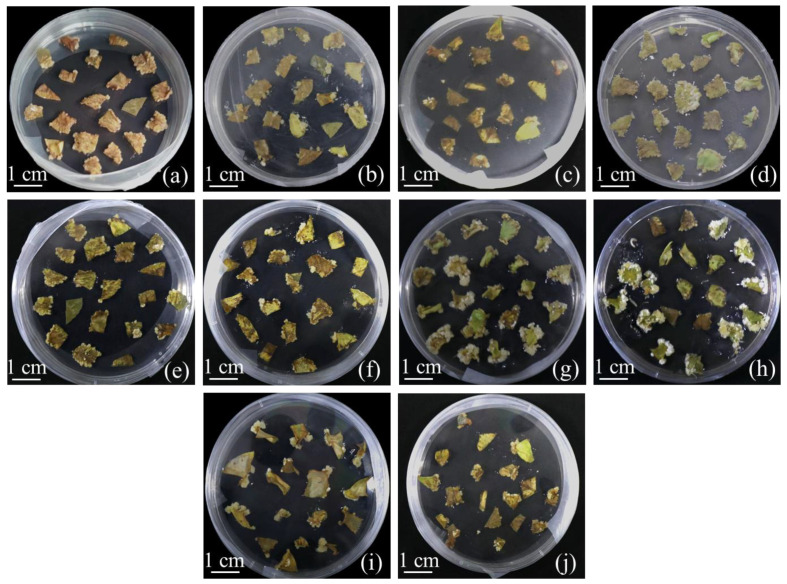
Calluses of different varieties of white clover at 20 days old emerged from the leaf wounds: (**a**) Barbzan callus; (**b**) Koala callus; (**c**) HaHnony callus; (**d**) Ladino callus; (**e**) Sulky callus; (**f**) Haifa callus; (**g**) Miracle callus; (**h**) Pixie callus; (**i**) Zapican callus; (**j**) MAG callus.

**Figure 2 ijms-24-11260-f002:**
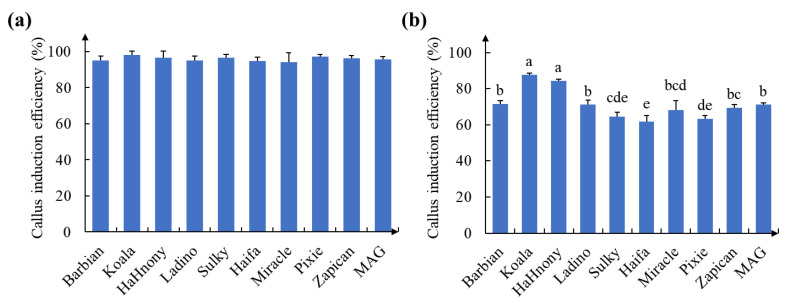
Callus initiation efficiency after 20 days of cultivation: (**a**) cotyledons of different varieties of white clover; (**b**) leaves of different varieties of white clover. Different letters indicate significant differences at *p* < 0.05.

**Figure 3 ijms-24-11260-f003:**
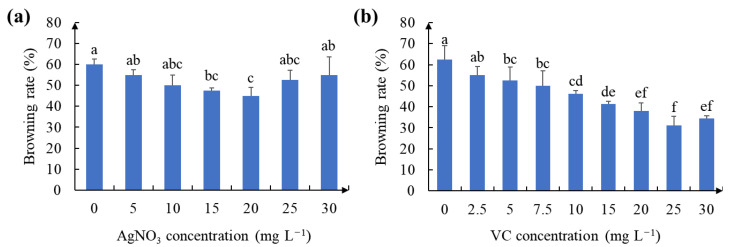
Effect of different antioxidant concentrations on the initiation of callus from leaf explant: (**a**) browning rate of callus on CCM medium with different quantities of AgNO_3_; (**b**) browning rate of callus on CCM medium with different quantities of VC. Different letters indicate significant differences at *p* < 0.05.

**Figure 4 ijms-24-11260-f004:**
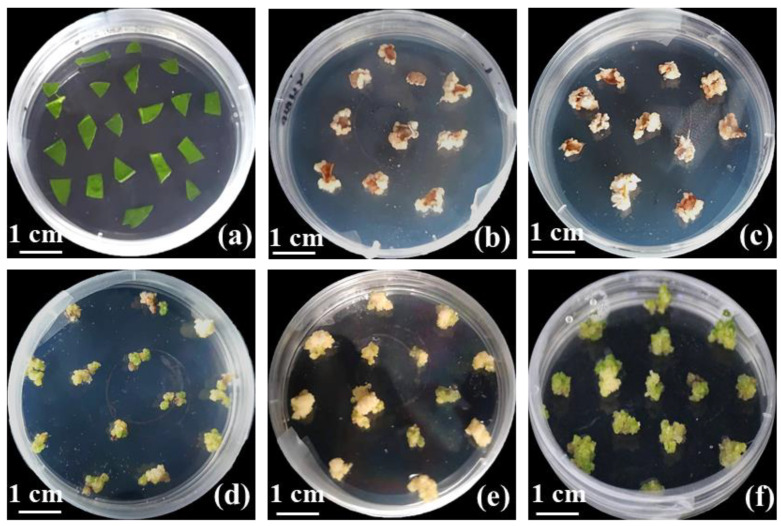
Effect of different antioxidant concentrations and hormone concentrations on browning and differentiation of calluses: (**a**) sterilized leaves on CCM; (**b**) calluses formed on CCM supplemented with 20 mg L^−1^ AgNO_3_; (**c**) calluses formed on CCM supplemented with 25 mg L^−1^ VC; (**d**) differentiation of embryogenic calluses on CDM1 containing 0.25 mg L^−1^ NAA; (**e**) differentiation of embryogenic calluses on CDM2 containing 1 mg L^−1^ KT; (**f**) differentiation of embryogenic calluses on CDM2 containing 1 mg L^−1^ 6-BA and 1 mg L^−1^ KT.

**Figure 5 ijms-24-11260-f005:**
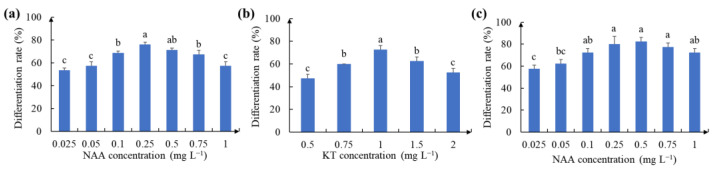
Effect of different hormone concentrations on the differentiation ability of leaf callus: (**a**) differentiation rates of callus on CDM1 medium with different quantities of NAA; (**b**) differentiation rates of callus on CDM2 medium with different quantities of KT; (**c**) differentiation rates of callus on CDM1 medium with 1 mg L^−1^ KT and different quantities of NAA. Different letters indicate significant differences at *p* < 0.05.

**Figure 6 ijms-24-11260-f006:**
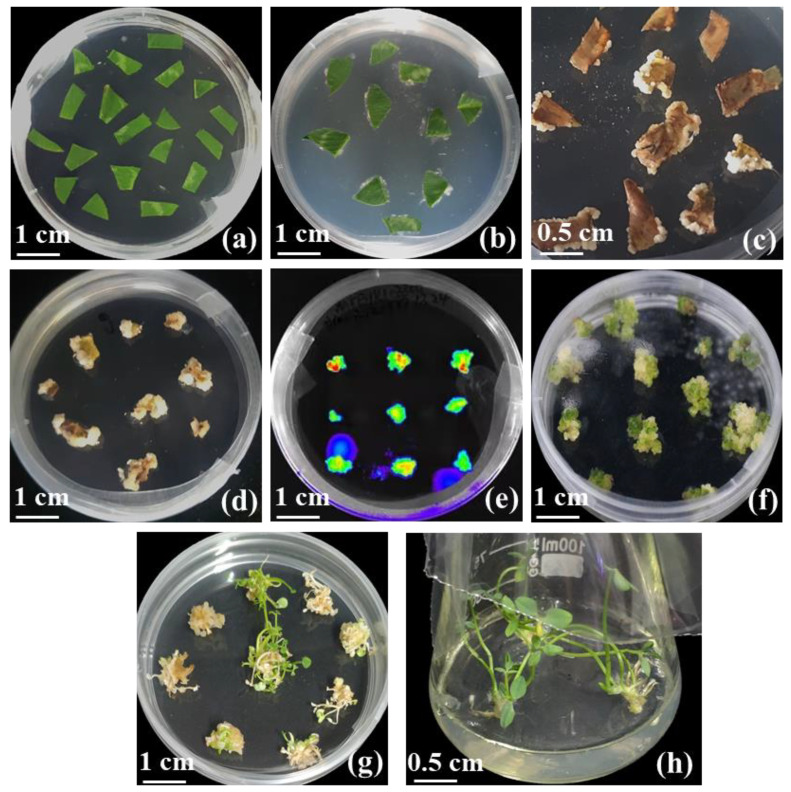
Procedure of *Agrobacterium tumefaciens*-mediated transformation of white clover: (**a**) sterilized leaves on CM; (**b**) leaves after 4 days of coculture with *Agrobacterium*; (**c**) 20-day-old calluses that emerged from the leaf wounds; (**d**) proliferation of transformed embryogenic callus on CDM containing 25 mg·L^−1^ glufosinate. (**e**) visualization of GFP fluorescence in embryogenic calluses; (**f**) calluses formed on CCM supplemented with 20 mg L^−1^ AgNO_3_; (**g**) proliferating glufosinate-resistant shoots after 60 days on CDM; (**h**) resistant plants were transplanted onto CRM-S for root development.

**Figure 7 ijms-24-11260-f007:**
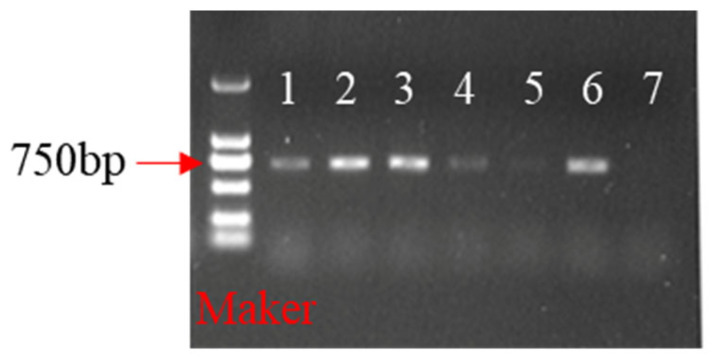
PCR amplification for the NPTII gene: 7—nontransformed white clover (control); 1–4, and 6—positive transgenic plants; 5—negative white clover.

## Data Availability

The data presented in this study are available upon request from the corresponding author.

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
