# Peer review of "Optimization of Protocols for the Induction of Callus and Plant Regeneration in White Clover (*Trifolium repens* L.)"

_ijms, 2023, doi:10.3390/ijms241411260_

Round 1

Reviewer 1 Report

Comment1: Why you only choose “Koala” as white clover leaf explants instead of selecting the highest and lowest varieties together to continue the following experiment? I don't believe it's quite convincing to choose just one variety.

Comment2: The CDM2 medium in Figure 4 is not clearly described above, which will cause confusion, please mark it above.

Comment3: The comparison between b and c in Figure 5 does not clearly show the difference that VC can reduce browning rate of embryonic calluses better than silver nitrate. Can you show a more obvious contrast figures ?

Author Response

Dear Reviewers:
I have put my response in the file, please see the attachment.

Reviewer 2 Report

1. The title itself should  be reflected on the parameters that were gathered.

2. Recast the title- since this is not really focusing on transformation, no data presented on transformation.

Minoe English editing needs to attend.

Author Response

(The authors gave the same response as above.)

Round 2

Reviewer 2 Report

This is better version